# Fluctuating Asymmetry in Ground Beetles (Coleoptera, Carabidae) and Conditions of Its Manifestation

**Sukhodolskaya Raisa [1],*, Saveliev Anatoliy [2] , Mukhametnabiev Timur [1] and Eremeeva Natalia [3]**

[1]  The Institute of Problems in Ecology, Tatarstan Academy of Sciences, 420087 Kazan, Russia; ctr_matrix@gmail.com
[2]  Department of Ecological Systems Modeling, Kazan (Volga river Region) Federal University, 420002 Kazan-2, Tatarstan, Russia; anatoly.saveliev.aka.saa@gmail.com
[3]  Institute in Biology, Ecology and Natural Resources, Kemerovo State University, 650000 Kemerovo City, Russia; neremeeva@mail.ru
*   Correspondence: sukhodolskayaraisa@gmail.com or ipen-anrt@mail.ru; Tel.: +7-9503152619

**Abstract:** Fluctuating asymmetry (FA) is used to reveal environmental or genetic stress, but the results of some studies are inconsistent. We aimed to give some explanations of possible controversial conclusions, when FA was employed. We measured FA (one dimensional and one meristic traits) in the recognized bioindicators—ground beetles (Coleoptera, Carabidae). Beetles were sampled in a vast area (four provinces of Russia with the spectrum of the studied sites, which differed in anthropogenic impact, vegetation, and landscape features). On the basis of such measurements (4673 specimen) we created a data base. Subsequent ANOVA showed, that FA was species-specific (out of six species investigated it was expressed in five ones), sex-biased (males had higher levels of FA), and were affected practically by all environmental factors. Besides significant species–sex and factors–sex interactions were found. So, when employing FA as an indicator of stress, overall biological and ecological variation in species-indicator must be investigated before. Sometimes FA (or its absence) may not be due to pollution or another disturbing factor, but be the result of the effect of unaccounted but FA determinative factors.

**Keywords:** inconsistency in fluctuating asymmetry studies; ground beetles; species-specificity; sex-specificity; environment factors specificity

## 1. Introduction

Fluctuating asymmetry (FA) is a random small deviation from perfect symmetry. It is regarded as an individual-based proxy of environmental and genetic stressors in a variety of taxa [1–5]. Both sides of bilateral traits develop under control of an identical genome. Then FA is assumed to determine the inability of organisms to suit their development against random perturbations. The latter is known as developmental instability (DI). Thereby FA is considered to be a mirror of the level of stress to which they are imposed (reviewed in [6,7]. A lot of papers were prepared about the FA–stress relationships, but it was shown that those associations were species-, population-, or trait-biased [8–10]. Those facts hamper FA use as indicator in evolutionary and conservation ecology and biology [11,12]. Inconsistency of those results can be explained by different factors. For example, when traits underwent large directional changes FA was higher [13]. Researchers found exterior and interior factors, including nutritional stress and lack of heterozygosity [14]. It is assumed that FA is affected by internal factors, i.e., genetics (inbreeding pressure and the disturbance of co-adapted gene complexes), and physiological

changes caused by non-native environmental shifts [15,16]. Lens et al. [11] studied the relationships between FA and inbreeding and argued whether FA can be considered as a "biomarker" for evaluating environmental and genetic stresses [12,17–22].

Carabid beetles are the large family, whose representatives are highly sensitive to environmental changes. Thereby they often are used as bioindicators [23–28]. Their population characters are often used to elicit environmental impact on biota and some ecogeographical rules [29–36]. FA is estimated in carabids populations also, but the results of such studies are sometimes contradictory [37–40].

The aim of our study was to examine:

(i)     Is FA species-specific in related species of ground beetles;
(ii)    Is FA sex-biased;
(iii)   How environmental factors affect FA in taken separately species.

## 2. Materials and Methods

Collection sites and sampling methods. The beetles from four large provinces were analyzed (Figure 1, Table 1). Wild specimens of ground beetles were sampled in different regions of Tatarstan Republic from 1996 to 2017. For the sake of this research, specimens from the other three provinces of Russia were transferred to us from carabidologists, who work in Perm, Kemerovo Universities and Visim Reserve. We measured those beetles ourselves (Table 2).

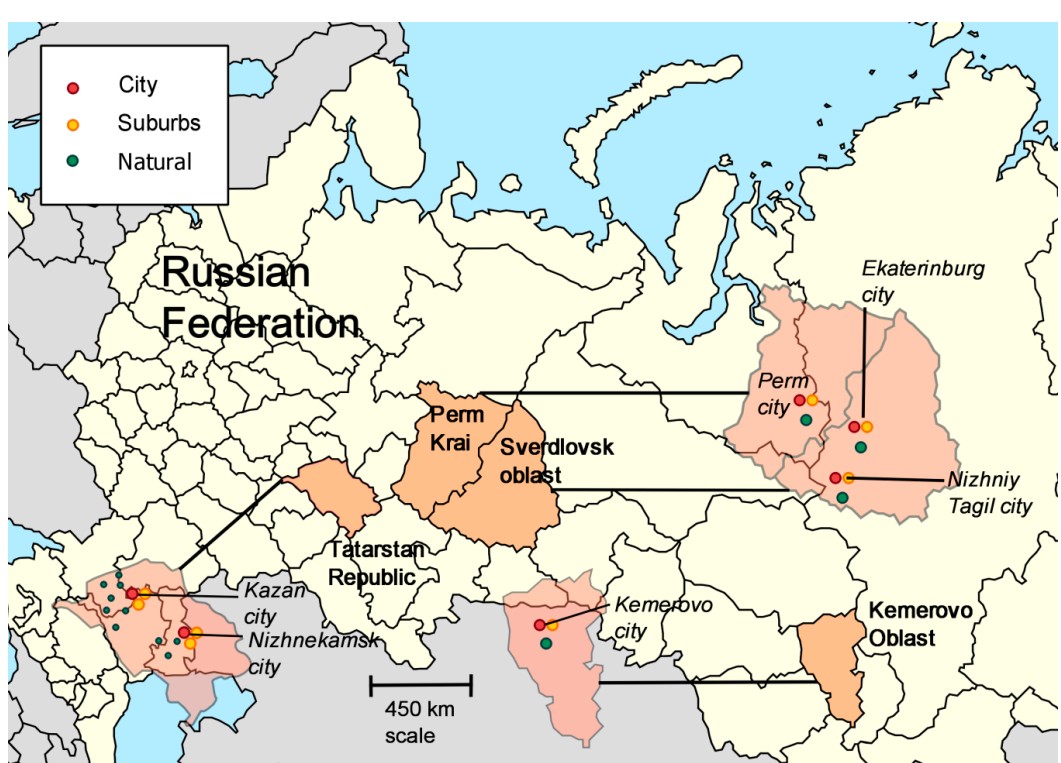

**Figure 1.** Sampling places.

**Table 1.** Sampling localities of carabids.

|   | Region | Latitude, °N | Longitude, °E |
|---|---|---|---|
| 1 | Kemerovo region | 54°56′ | 87°14′ |
| 2 | Tatarstan Republic | 55°47′ | 49°06′ |
| 3 | Perm kray | 57° 01′ | 57°9′ |
| 4 | Sverdlovsk region | 58°42′ | 61°20′ |

**Table 2.** Studied species of carabids and data analyzed.

|  | Species | Number of Sites | Sample Size |
|---|---|---|---|
| 1 | *C. aeruginosus* | 3 | 528 |
| 2 | *C. cancellatus* | 4 | 774 |
| 3 | *C. granulatus* | 4 | 865 |
| 4 | *P. melanarius* | 5 | 470 |
| 5 | *P. niger* | 2 | 59 |
| 6 | *P. oblongopunctatus* | 2 | 305 |
| 7 | *Poec. cupreus* | 10 | 1672 |

In Tatarstan we sampled ground beetles in their common habitats, which were mainly similar between studied regions for all studied species as well as our colleagues from other provinces. Beetles in every province were pitfall trapped at the territories with differing urbanization press—in cities, suburbs, arable lands and natural biotopes. We summarized the details on sample sizes in Table 3.

**Table 3.** Number of measured specimen.

| Species | Kemerovo | | | Tatarstan | | | | Perm | | | Sverdlovsk | | |
|---|---|---|---|---|---|---|---|---|---|---|---|---|---|
| | City | Suburbs | Natural | City | Suburbs | Natural | Agro | City | Suburbs | Natural | City | Suburbs | Natural |
| *C. aeruginosus* | 210 | 80 | 238 | - | - | - | - | - | - | - | - | - | - |
| *C. cancellatus* | - | - | - | 121 | 48 | 205 | - | 30 | 81 | 79 | 35 | 84 | 89 |
| *C. granulatus* | 85 | 73 | 119 | 79 | 75 | 186 | - | 33 | 31 | 54 | 48 | 37 | 40 |
| *P. melanarius* | - | - | - | 49 | 32 | 51 | - | 44 | 52 | 51 | 30 | 68 | 93 |
| *P. niger* | 30 | - | - | 29 | - | - | - | - | - | - | - | - | - |
| *P. oblongopunctatus* | - | - | - | 29 | 30 | 49 | - | 31 | 32 | 30 | - | - | - |
| *Poec. cupreus* | 48 | 67 | 70 | 38 | 161 | 517 | 515 | - | - | - | 65 | 92 | 104 |

Study organisms. We analyzed six ground beetles species: *Carabus aeruginosus* (Fischer von Waldheim, 1823), *Carabus (Carabus) granulatus* Linnaeus 1758, *Carabus (Tachypus) cancellatus* Illiger 1798, *Pterostichus melanarius* Illiger 1798, *Pterostichus niger* Schaller 1783, *Pterostichus oblongopunctatus* Fabricius 1787, and *Poecilus cupreus* Linnaeus 1758. All of them (except *C. aeruginosus*) are widespread in Palearctic, euribionts, active predators and mesophilous. *C. aeruginosus* is a Siberian species.

Morphometric analysis. All measurements were made with a Leitz RS stereoscopic microscope at a magnification of 10 diameters. We used a calibrated ocular grid. Its scale interval was 0.1 mm. For each of specimens (except *P. oblongopunctatus*) we measured the right and left elytra width (further dimensional trait). Besides dimensional trait we analyzed meristic traits and counted the number of tubercles in the first line near medial ridge of the scutellum (in *C. granulatus* and *C. cancellatus*), the number of spots (in *P. oblongopunctatus*), and the number of furrows on the left and right elytra (in *P. melanarius, P. niger,* and *Poec. cupreus*). *C. aeruginosus* has no such meristic traits, so data on it was absent in certain tables in Results and Supplement.

Statistical analysis. For each specimen we calculated fluctuating asymmetry (FA) index (FA = |R − L|/(R + L)/2, where R is the value of the trait at the right elytra, while L denotes the value of the trait at the left elytra.

We used a GLM to recognize what environmental factors affected FA and if FA values were sex-specific: DimensionalAsim ~ fSpecies + fProvince + fAnthropogen + fSex (here sign "~" means "depend on", and sign "+" means sum of effects) and MeristicAsim ~ 0 + fSpecies + fProvince + fAnthropogen + fSex. We did not include "biotope" variable into the models, because not all biotopes were presented in all locations, which might lead to an unbalanced design.

Those models and their ANOVA estimates variables (species, province, etc.) effects on FA variation. Model coefficients interpretation for these kinds of models is difficult. Since we used categorical independent variables, to avoid over-parameterization when using dummy variables, one of the variable level become non-estimable, and it's impact is included in intercept. In our models results intercept was negative, because of the data structure (independent variables correlation, etc.).

Besides, we could not estimate the significance for a specific variable value: we needed to compare it with zero, but could compare with some non-zero "base level", defined by effects, included in intercept. Therefore, to estimate influence on FA of specific values of variable (specific species for fSpecies, etc.) we needed a model without intercept with single variable (assuming the impact of other factors was random and neglectable). For each species the model was as follows formula = lm (DimAsim ~ 0 + fSpec, data = p). For example, to estimate the effect for each of the species the model was as follows: DimAsim ~ 0 + fSpec, where 0 meant the intercept. All results of those models are given in the Supplement.

## 3. Results

Such factors as "Species", "Province", and "Anthropogene" affected FA in the dimensional trait in carabids (Tables 4 and 5). In all the tables the following significance level of results is accepted: '***' 0.001 '**' 0.01 '*' 0.05 '.' 0.1 ' ' —data is absent. *C. cancellatus* and *P. cupreus* and habitation in cities contributed significantly into FA variation.

**Table 4.** ANOVA results of environmental factors effect on fluctuating asymmetry (FA) in the dimensional trait in ground beetles.

| Source | D f | Sum_ of Sq | R_S_S | A_I_C | F-value | Pr_(>F) | |
|---|---|---|---|---|---|---|---|
| <none> | | | 8.036 | −18194 | | | |
| Species | 5 | 0.093 | 8.129 | −18169 | 7.055 | 0.000 | *** |
| Province | 2 | 0.020 | 8.056 | −18190 | 3.822 | 0.022 | * |
| Anthropogene | 3 | 0.046 | 8.082 | −18183 | 5.856 | 0.001 | *** |
| Sex | 1 | 0.000 | 8.036 | −18196 | 0.001 | 0.973 | |

**Table 5.** GLM results of environmental factors effect on FA in the dimensional trait in ground beetles.

| Source | | ESTIMATE | STD. Error | t-value | Pr (>\|t\|) | |
|---|---|---|---|---|---|---|
| | (Intercept) | −0.010 | 0.006 | −1.841 | 0.066 | . |
| | *C. cancellatus* | 0.024 | 0.009 | 2.566 | 0.010 | * |
| | *C. granulatus* | 0.014 | 0.011 | 1.217 | 0.224 | |
| Species | *P. cupreus* | 0.032 | 0.008 | 3.814 | 0.000 | *** |
| | *P. melanarius* | 0.003 | 0.011 | 0.323 | 0.747 | |
| | *P. niger* | −0.012 | 0.013 | −0.938 | 0.348 | |
| Province | Sverdlovsk oblast | 0.020 | 0.012 | 1.581 | 0.114 | |
| | Tatarstan republic | −0.003 | 0.009 | −0.298 | 0.766 | |
| | city | 0.019 | 0.005 | 3.852 | 0.000 | *** |
| Anthropogene | natural | 0.008 | 0.010 | 0.840 | 0.401 | |
| | suburb | 0.013 | 0.007 | 1.865 | 0.062 | . |
| Sex | males | 0.000 | 0.002 | −0.034 | 0.973 | |

In the meristic trait the results were similar, but contribution of factors differed: *Poec. Cupreus* and *P. niger* effects were significant. Several factors highly significantly affected FA—all provinces and habitation in cities, suburbs, and natural biotopes (Tables S1 and S2). The real contribution of separate variables was difficult, because some variable impact was included into the intercept. However, another modeling gave the following results (Tables S3–S6): in the dimensional trait FA was affected by province and biotope, but in the meristic trait by anthropogene.

FA in the dimensional trait differed in different provinces being slightly lower in beetles from Kemerovo oblast´. In the meristic trait FA was significantly higher, especially in Perm´kruy beetles (Figure 2). FA was not sex-biased, being almost the same in females and males (Figure 3). However, again FA in the dimensional trait was lower than in the meristic.

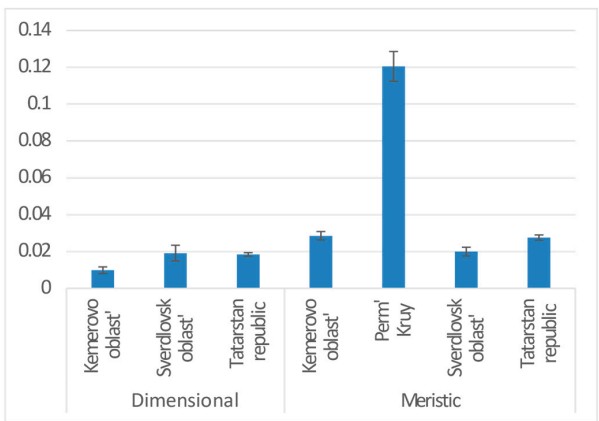

**Figure 2.** FA-values in carabids in different provinces of Russia.

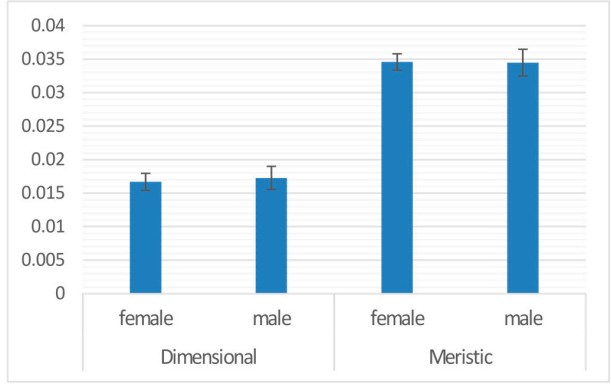

**Figure 3.** FA values in carabids.

FA did not differ significantly in beetles inhabiting cities, agrolandscapes, and natural habitats, but were lower in ground beetles in suburbs. Related to the meristic trait results differed: FA was low in beetles in agrolandscape and significantly higher in the beetles at natural territories and cities and suburbs (Figure 4).

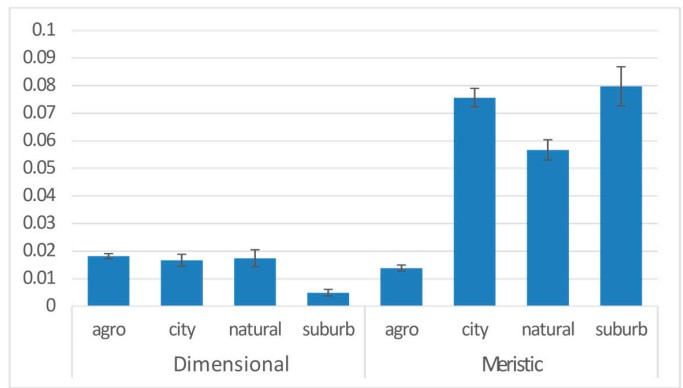

**Figure 4.** F-values in carabids at territories with different level of urbanization.

Linear models use provided us with more concrete results, because the only one external or internal factor impact was analyzed in each of them (Tables S7–S12). All studied species (except *P. niger*) showed FA. FA was recorded in females and males as well. Modeling species–sex interaction showed, that it was highly significant in all species studied and in males FA was higher (except *C. cancellatus*).

FA in meristic trait was higher when tubercules (*C. granulatus* and *C. cancellatus*) or spots (*P. oblongopunctatus*) had been counted. Sex impact on FA was significant also, especially in males. Sex–species interaction was significant too, but in three out of five studied species, FA was higher in males than in females. So, in all treated species FA was sex-biased, i.e., in females and males its expression differed.

When we modeled environmental factors on FA variation in taken separately species, *Poec. cupreus*, results were as follows: habitat type, anthropogene (city or agrilandscape), and landscape features (sampling in floodplain or plakor) affected FA expression in studied species (Tables S13–S27).

Overwhelming majority of studied factors affected FA in that species: habitat type (except spring wheat), and anthropogene (sampling in city or agricultural landscape). Besides all effects were sex-specific, i.e., FA was expressing not similarly in females and males. Some differences occurred in FA in dimensional and meristic traits. So, the significance of the factor effect on FA in the meristic trait was lower in the cases with winter wheat, barley, carrot, winter wheat in females, lucerne, oat, and pea in males. On the contrary, in carrot FA in the dimensional trait it was not registered, but in meristic it occurred.

## 4. Discussion

Results show that FA expression depended on plenty of factors. First, it was sex-specific. Even closely related species showed different sensitivity to the same factor. In our study there was *P. melanarius* and *P. niger*, the latter being useless as an indicator with FA employment. Sex-specificity of FA expression was important too. If the sex-ration in the sample was not in equilibrium, viz. 1:1, then predominance of one or another sex led to the distorted results when FA estimating, since impact of the same factor on stability of development in females and males differed. These could explain some inconsistencies in studies on ground beetles. Our results concurred with other studies. Impact of environmental factors on FA expression profoundly was shown in H. Benitez studies. In the perennial agro-ecosystem (more stable environment) FA of carabids shape was lower than in annual arable lands (severe environment) [41]. This study was similar with results, which suggested that FA could estimate habitat quality when comparing conventional and organic farming [40]. Significant differences in FA values were shown in ground beetle *Ceroglossus chilensis* in 6–7 (first tree thinning) and 10 years old plantations (commercial thinning) [42].

In our study FA in ground beetles in the agrolandscape did not differ significantly from the FA in the beetles from the cities and natural habitats. Perhaps, the case when the data set combines all species studied, masked some effects. This was due to the fact that in our dataset the only one species from the agrolandscape, *Poec. cupreus,* was present. This species is common in agrolandscapes and supports the fitness through body size and shape variation in different crops [43]. The same fact could explain the low FA level in city beetles. All studies species were generalists and adapted quite well to the urbanized environment gradient [29]. Therefore the level of FA in such an environment did not differ greatly from natural ones. When investigating the body shape of two populations of *Ceroglossus chilensis* (Eschscholtz), there has been shown that FA was higher in the populations exposed to the severe environment—in drier areas at Andes Foothills, comparing with populations dwelling at more humid Coast Range [44]. Environmental factors, such as nutritional stress, temperature, chemical pollution and population density, cause stress during development can lead to increased FA in Coleoptera [42,45,46]. Temperature effect on development was shown in other studies: body size in species, mentioned in our investigation, decreased towards high latitudes [33]. Perhaps significantly higher FA in Sverdlovsk and Tatarstan beetles dimensional trait was determined by a more northern location of those provinces if comparing with Kemerovo. A very high level of FA in the meristic trait in Perm´ kruy was determined by the cold climate in that region too.

Species-biased FA was shown in the study with the conclusion that farming practice affected FA in specialists but not generalists [40]. Sex-specific FA was shown for ground beetles near the Danish city—in females the FA level was higher than in males [38]. Body condition, which correlated negatively

with FA, was higher in females [39], but in the whole the authors concluded that ground beetles sometimes could not be used as indicators of environmental quality by FA employing. Nevertheless the same authors found a significant negative correlation between condition and asymmetry for *C. nemoralis* and *N. brevicollis* in the suburban as well as urban forest fragments [39].

Perhaps, it was that case, when the authors sampled the wrong object. The same comments can be given in relation to other studies, when in terms of verification of the center–periphery hypothesis, researchers predicted higher levels of wing asymmetry at the margins of area. However, the FA level in *Drosophila* species was similar along the whole area [47]. Organic and integrated pest management did not affect FA level in *P. melanarius*, so the authors suggested that FA could not be used as a bioindicator [37]. Beetles exposure to Cd and Zn did not change the FA level in those objects also [48].

Finally one comment would be given about biology of traits employed to FA estimation. In all cases we studied FA in the dimensional trait was smaller than in meristic. Perhaps it is due to higher variability in phenotypic traits estimated by counting, not measuring. Additional research is needed in this field, which will highlight perspectives FA tools in environment quality evaluation.

## 5. Conclusions

Our study was planned in such a way that we could clarify different factors impact on FA manifestation in ground beetles. We revealed that FA in carabids is species-specific—in some species it appeared, and in some it did not. So, when employing FA to register any environmental stress, some species of carabids could not be used as indicators. Secondly, FA was sex-biased. It was not surprising because of the different roles males and females play. Then when estimating FA in the carabids sample, attention should be paid for the sex-ratio, as many males in ground beetles show higher FA levels, when interacting with environmental factors, especially with urbanization. Finally, when employing FA as an indicator of stress, overall biological and ecological variation in the species-indicator must be investigated before.

**Supplementary Materials:** The following are available online at http://www.mdpi.com/2073-8994/11/12/1475/s1.

**Author Contributions:** Author Contributions: Conceptualization, S.R.; Methodology, S.R., M.T.; Software, S.A.; Validation, E.N.; Formal analysis, M.T., E.N.; Investigation, M.T., S.R.; Resources, S.R.; Data curation, S.A., E.N.; Writing, S.R.; Visualization, S.A.

**Funding:** This research received no external funding.

**Acknowledgments:** We thank our colleagues from Perm State University, Visim State Reserve for beetles samples, presented for measurements. We thank the staff of Laboratory of Biomonitoring of Tatarstan Academy of Sciences for beetles sampling in Tatarstan Republic and the high quality assistance in their measurements.

**Conflicts of Interest:** The authors declare no conflict of interest.

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
