# Peer review of "Fluctuating Asymmetry in Ground Beetles (Coleoptera, Carabidae) and Conditions of Its Manifestation"

_symmetry, doi:10.3390/sym11121475_

Round 1

Reviewer 1 Report

This manuscript is dealing with the fluctuating asymmetry of six ground beetle species. The topic of the manuscript is very interesting, since fluctuating asymmetry seems to be an individual-based proxy of environmental stressors. Furthermore, the studied ground beetles play a vital role in the food chain, and have important ecosystem services; therefore, their ecological responses to environmental disturbances are key issue for both managers and conservationists. The manuscript is generally well written, the analyses are well explained and straightforward. There are, however, some essential points that need to be addressed before making a decision on publication of the manuscript.

The used fluctuating asymmetry (FA) index (FA = |R – L| / (R + L) / 2) is recommended to use only where clear evidence exists of a size dependence of |R – L| among individuals within a sample (see Palmer, A.R. 1994. Fluctuating asymmetry analyses: A primer, In: Markow, T.A. (ed.), Developmental Instability: Its Origins and Evolutionary Implications. Kluwer, Dordrecht. pp. 335–364.). Therefore, the size dependence of FA should be tested by linear regression of the absolute deviation of R–L (|R–L|) values against an independent measure of body size. Instead of several separate linear models, a general linear model (GLM) treating species, gender and the studied factors as fixed factors (and their interactions) would be a more appropriate method to test comprehensively differences in FA. Please reanalyse your data by GLM. The Discussion section is superficial, it only lists the previous studies and their results. There is no discussion of what factors may have caused the showed differences in FA.

Minor comments

L. 17. and through the manuscript. Please change "Ground Beetles" to "ground beetles" through the manuscript.

L. 55. It may be worth to cite a recent paper on this topic (Basic and Applied Ecology 7 (2006) 472-482).

L. 69-70. Please add the exact number of studied individuals.

L. 86. Please change "merestic" to "meristic".

L. 89. Please change "right elytra left elytra" to "left elytra".

L. 92. Please change "merictic" to "meristic".

L. 106. Please change "exrensively" to "extensively".

L. 180. Please delete "c-".

Author Response

We are thankful for our reviewers for their attention to our work. Some their comments were taken into account when MS revising. Responses to other notes are presented in corresponding files in color letters.

Reviewer 2 Report

This paper presents the results of FA measurements in ground beetles in several provinces in Russia. The data set is very large (> 4000 individuals) and two different types of traits were measured (a metric and meristic trait). It is not made clear why this is done, nor are specific comparisons presented in the paper.

Overall the writing of this paper needs improvement, especially the language needs specific attention. I strongly recommend to involve a scientist fluent in English to revise the paper.

While the sample size of the study is large and thus may shed light on variation in FA at different levels, the analysis is fairly poor. In the methods section the authors present some code they used in (I presume) R. People not familiar with this package will not understand what models are fitted. I am familiar with R, and can see that the ANOVA models are not really ANOVA models because no intercept is estimated. Thus, tests do not make any comparisons among groups. In addition, tables with only stars (***) indicating significance are not very helpfull to understand differences and so on.

The analyses should also take into account differences between provinces and perhaps also variation among sites within provinces. The reader is not given a good view on how the sampling was actually done. It is, for example, not clear how the different types of sample sites are distributed within and among provinces. The authors may also want to include both measures of FA in the analysis and estimate the correlation among them (i.e., a test for the presence of an individual asymmetry parameter)

The authors also need to decide of they want to treat species as a fixed or random effect in their models. The key question is 'do you want to compare patterns among the different species in a two-by-two fashion or do you want to estimate between-species variation'. Personally I would go for the second option, but the authors may have good reasons to treat speacies as a fixed factor.

Another point that requires attention is what to do with a possible 'genus' effect. 3 genera are studied, and the data within a genus may be more similar that across genera.

A weakness of the paper is that no repeated measurements were available. Stricktly speaking, one does not know if the FA values are real or driven by measurement error. It is also not clear of DA is present or not. I understand of course that given the very large sample size, repeated measurements for all individuals are difficult to obtain. I would suggest to do repeated measurements for a subset of individual to rule out that much of the FA variation is actually ME.

I would also like to see some graphs which summarize the main results that lead to the major conclusions of the paper.

So overall I think the dataset has merit to become a nice contribution to the literature, but a major revision of the text, the analyses and the presentation of the results is urgently needed

Author Response

(The authors gave the same response as above.)

Round 2

Reviewer 1 Report

This manuscript is dealing with the fluctuating asymmetry of six ground beetle species. Majority of points emerged during the previous evaluation process had been satisfactory addressed. However, there are still some essential points that need to be addressed before making a decision on publication of the manuscript.

If several separate linear models are used, the likelihood of incorrectly rejecting a null hypothesis (i.e. making a Type I error) increases. Therefore, I still suggest to test differences in FA by general linear model (GLM) or by generalized linear mixed model (GLMM) treating species, gender and the studied factors as fixed factors (and their interactions).

Please rewrite the currently rather superficial discussion. Please focus on the environmental factors which can be affected FA.

Reviewer 2 Report

The authors have submitted a revision of their paper and made some improvements with respect to the sampling sites and sample sizes.

With respect to my suggestions towards the statistical analyses and presentation of the results, the authors seem a bit agitated and basically neglected my suggestions. I realize that I basically requested a complete reanalysis of their data and a thorough rethinking of the hypothesis of interest. I also wanted to see graphs with results and not just stars indicating statistical significance. I made these suggestions in order to improve the quality and merit of this interesting dataset, not to harass the authors. I still consider the analyses presented as very poor and not of great interest to potential readers. There are no clear hypotheses behind it and the analysis is basically only a huge set of one-sample t-tests even though the authors claim to have done ANOVA's. Linear models in general and ANOVA in particular are usually meaningless when no intercept is modeled.

I thus consider this revision as only a limited improvement and refer to my first evaluation for further directions.

Round 3

Reviewer 1 Report

This manuscript is dealing with the fluctuating asymmetry of six ground beetle species. All points emerged during the previous evaluation processes were addressed in the revised manuscript. There are, however, some minor technical points that should be considered before publishing this manuscript.

Please change "Ground Beetles" to "ground beetles" through the manuscript. Figures 2-4: please use decimal point instead of comma. L. 195. Please change "at the at the" to "at the".

Reviewer 2 Report

I am happy to see the revisions made, it adds a lot to the clarity of the paper and the main results that are presented. Some clarifications may be helpfull:

1) why is biotope added to the model? I am pretty sure that not all biotopes are present in all locations and levels of antropogenic disturbance. A very unbalanced design may lead to spurious results, so I would suggest to leave it out (even though it is statistically significant)

2) how can the intercept of your model be negative. This is simply not possible given the way FA is calculated. There thus must be somethin g(hopefully minor) wrong with the statistical model or the description of how FA was calculated
